# Hyperpolarized Metabolic MRI—Acquisition, Reconstruction, and Analysis Methods

**DOI:** 10.3390/metabo11060386

**Published:** 2021-06-14

**Authors:** Peder Eric Zufall Larson, Jeremy W. Gordon

**Affiliations:** 1Department of Radiology and Biomedical Imaging, University of California, San Francisco, CA 94143, USA; jeremy.gordon@ucsf.edu; 2UC Berkeley-UCSF Graduate Program in Bioengineering, University of California, San Francisco, CA 94143, USA; 3UC Berkeley-UCSF Graduate Program in Bioengineering, University of California, Berkeley, CA 94143, USA

**Keywords:** hyperpolarized carbon-13 MRI, metabolic imaging, spectral-spatial RF pulses, variable flip angles, fast spectroscopic imaging, metabolite-specific imaging, kinetic modeling

## Abstract

Hyperpolarized metabolic MRI with ${}^{13}$C-labeled agents has emerged as a powerful technique for in vivo assessments of real-time metabolism that can be used across scales of cells, tissue slices, animal models, and human subjects. Hyperpolarized contrast agents have unique properties compared to conventional MRI scanning and MRI contrast agents that require specialized imaging methods. Hyperpolarized contrast agents have a limited amount of available signal, irreversible decay back to thermal equilibrium, bolus injection and perfusion kinetics, cellular uptake and metabolic conversion kinetics, and frequency shifts between metabolites. This article describes state-of-the-art methods for hyperpolarized metabolic MRI, summarizing data acquisition, reconstruction, and analysis methods in order to guide the design and execution of studies.

## 1. Introduction

MRI with hyperpolarized carbon-13 agents, or hyperpolarized (HP) ${}^{13}$C MRI, has created a new type of non-invasive, in vivo metabolic imaging that can be applied in cell, animal, and human studies [1]. The use of ${}^{13}$C-labeled agents, primarily [1-${}^{13}$C]pyruvate, enables monitoring of key metabolic pathways in near real-time with the ability to image substrate and products based on their chemical shift [2]. [1-${}^{13}$C]pyruvate has been especially successful because its properties are well suited for hyperpolarized imaging studies (e.g., long $T_{1}$, high concentration preparations) and because of pyruvate’s critical position in glycolysis, where it can either be used in TCA cycle metabolism or converted to lactate. Many other ${}^{13}$C-labeled agents have been successfully developed in animal studies, including ${}^{13}$C-urea as a measure of perfusion [3], ${}^{13}$C-fumarate as a measure of necrosis [4], ${}^{13}$C-alpha ketoglutarate as a measure of IDH status [5], ${}^{13}$C-bicarbonate as a measure of extracellular pH [6], ${}^{13}$C-butyrate [7] and ${}^{13}$C-acetate [8] as a measure of fatty acid metabolism, ${}^{13}$C-dehydroascorbate as a measure of redox potential [9], and many more [10].

However, the hyperpolarized state is transient and relaxes to thermal Boltzmann equilibrium with time constant $T_{1}$. This $T_{1}$ relaxation is primarily a function of $B_{0}$ field strength and the local chemical environment. In general, nuclei with directly attached protons relax via dipole–dipole interactions and have $T_{1}$ times that are too short (<10 s) to be viable for HP studies [11]. However, the majority of compounds studied via dissolution DNP are carbonyls, alkenes, and other species that have no directly attached protons. These compounds tend to have long (>30 s) $T_{1}$ times and primarily relax via chemical shift anisotropy (CSA), with relaxation rates (1/$T_{1}$) proportional to $B_{0}^{2}$ [12]. For example, [1-${}^{13}$C]pyruvate, the most widely studied probe to date, has a $T_{1}$ in solution of 67 s at 3 T and 44 s at 14.1 T [10]. This $T_{1}$ decay, as well as the rapid bolus injection and metabolic conversion (via ${}^{13}$C label exchange), necessitates dedicated spectroscopic imaging techniques that can efficiently capture rapidly evolving signals within 1–2 min following injection. These concerns may be mitigated in the future by exciting work on long-lived spin states that offer the possibility to significantly extend the hyperpolarized lifetime, greatly alleviating this limitation [13].

This review will discuss the current state-of-the-art methods for hyperpolarized metabolic MRI, summarizing data acquisition, reconstruction, and analysis methods in order to guide the design and execution of studies.

## 2. Pulse Sequence Methods

Data acquisition strategies in HP MRI experiments must account for multiple chemical shifts, efficiently utilize the non-renewable HP magnetization, and acquire data quickly relative to metabolism and relaxation decay processes. Studies of metabolically active HP molecules require spectral encoding to separate metabolites, requiring pulse sequences to efficiently encode up to 5D data (3 spatial + 1 spectral + 1 temporal). Furthermore, RF pulses and experiment timing must efficiently sample the non-renewable HP magnetization, both for the original substrate as well as metabolic products. This section will describe state-of-the-art pulse sequences that have been used for hyperpolarized ${}^{13}$C MRI and will discuss their relative tradeoffs and benefits.

### 2.1. RF Excitation and Imaging Timing

The dissolution dynamic nuclear polarization process provides more than four orders of magnitude increase to nuclear polarization in the liquid state [14]. This transient increase overcomes the small thermal equilibrium magnetization—on the order of parts per million at clinical field strengths and physiologic temperature—and enables dynamic imaging of ${}^{13}$C-labeled compounds. However, this has significant consequences for the design of hyperpolarized studies, particularly in regard to the total imaging time and choice of flip angles.

In ${}^{1}$H MRI, the flip angle is chosen to provide a desired contrast or to maximize signal (i.e., Ernst angle for a spoiled gradient echo). In hyperpolarized MRI the choice of flip angle has a much greater impact, as it determines not only the overall signal-to-noise ratio (SNR) but also the total imaging time of the acquisition. Hyperpolarized spins decay to thermal equilibrium once they are removed from the polarizer. Implicit within this, however, is that each RF pulse consumes some of the finite, non-recoverable hyperpolarized magnetization.

There are fundamental tradeoffs between SNR and total imaging time (e.g., the ability to resolve kinetics) when choosing flip angles. Relatively small flip angles preserve Mz, allowing for longer experiment times, but provide smaller Mxy and a smaller signal and limit the spatial resolution. In contrast, a larger flip angle initially leads to a larger Mxy, and thus more signal, but because there is no recovery, the rapid consumption of Mz magnetization shortens and limits the opportunity to observe the kinetics.

However, most hyperpolarized ${}^{13}$C studies employ a substrate (such as [1-${}^{13}$C]pyruvate) that is metabolically active. Metabolites are generated through enzymatic conversion or label exchange from the hyperpolarized substrate and further spread out the hyperpolarized magnetization amongst multiple compounds. Given the presence of metabolic conversion, the frequency response of RF pulses can be used to modulate the flip angles for different compounds, improving the SNR in dynamic hyperpolarized experiments. A number of flip angle schemes [15,16,17,18,19] have been proposed that vary the flip angle across frequency and/or time to more efficiently sample the magnetization throughout the acquisition. One common way to approach this would be to apply a lower flip angle to the injected substrate, which is typically present at a higher concentration and has higher SNR than downstream metabolites that would be given a larger flip angle to boost their SNR, illustrated in Figure 1 and simulated using a precursor-product model discussed further in Section 3.1. Utilizing the same flip angle for both substrate and product results in excess substrate SNR and an overall reduction in product SNR because of the rapid drop in the substrate Mz. Using a multiband (or metabolite-specific) flip angle—lower excitation for substrate and higher for metabolite—results in reduced usage of the substrate hyperpolarization Mz, providing more product magnetization while still providing sufficient SNR for both throughout the acquisition.

The flip angle can also be varied through time, with schemes developed to either maximize the metabolite signal or provide a uniform signal throughout the acquisition [17,19], though it is important to note that these approaches tend to be more sensitive to errors in RF transmit power and variations in bolus delivery and perfusion [20,21].

The key takeaway here is that RF pulses and metabolism will further utilize the non-recoverable magnetization, and the tradeoffs between SNR for a single timepoint and total SNR throughout the entire acquisition must be taken into account. This implies that signal averaging or experiments that require a 90${}^{\circ }$ excitation are generally incompatible in hyperpolarized experiments. Refocused sequences that utilize high flip angles, such as fast spin echo [22] or steady-state free-precession (SSFP) [23,24,25], can be adapted for hyperpolarized studies, but care must be taken to properly calibrate the RF power.

### 2.2. Data Acquisition and Reconstruction Strategies

An efficient RF excitation and imaging timing strategy must be combined with data acquisition and reconstruction strategies that can rapidly encode up to 5D data (3 spatial + 1 spectral + 1 temporal). Spectral encoding is particularly important for assessing metabolism with HP ${}^{13}$C MRI in order to image both substrates and metabolic products, while temporal encoding allows for measurement of metabolism kinetics.

#### 2.2.1. Magnetic Resonance Spectroscopic Imaging (MRSI)

Magnetic resonance spectroscopic imaging (MRSI) strategies provide both spectral and spatial encoding for HP ${}^{13}$C studies. These techniques have the advantage that they provide spatial localization as well as a continuous spectrum, which can be analyzed for a large range of resonance frequencies. This makes MRSI very robust and the go-to method for exploratory HP studies when the number of resonances and their relative chemical shifts are unknown. These sequences have been particularly effective when paired with acceleration strategies that can increase the spatiotemporal resolution to more efficiently capture HP signals during their limited signal lifetime and rapid dynamics [26,27,28].

The most straightforward method for hyperpolarized ${}^{13}$C studies is 1D spectroscopy, which acquires a non-localized spectrum from a single slice or a region limited by the sensitivity of the receive coil (Figure 2 top). This approach provides spectroscopic data with a high spectral bandwidth and spectral resolution, with the capability for sub-second temporal resolution to capture rapid enzyme kinetics. This approach is well-suited for hyperpolarized substrates with complex resonance frequency patterns in the spectrum and/or a broad chemical shift range, for example [2-${}^{13}$C]pyruvate [29,30] or [U-${}^{2}$H, U-${}^{13}$C]glucose [31], or when spatial localization is not crucial and only global changes are expected.

For non-hyperpolarized MRSI studies (1H, 31P, etc), phase-encoded chemical shift imaging (CSI) is often used because it provides a large spectral bandwidth and high spectral resolution along with spatial encoding (Figure 2 middle). However, the main challenge in performing hyperpolarized MRSI is imaging speed, and CSI is quite slow because it is a pure phase-encoded sequence. Similar to 1D spectroscopy, phase-encoded CSI is most well-suited for pre-clinical studies with small FOVs [32,33] or for substrates with complicated spectra or large chemical shift dispersion where high spectral resolution and a large spectral bandwidth are needed, as its poor temporal resolution precludes volumetric coverage and can hamper measurements of metabolic conversion.

Fast spectroscopic imaging techniques employing multi-echo readouts during acquisition can greatly reduce the scan time for HP experiments compared to phase-encoded CSI (Figure 2 bottom). Joint spatial and spectral encoding is accomplished by traversing k-space at multiple TEs, shifted in time by ΔTE (Figure 2 bottom). The spectral and spatial encoding for rapid MRSI techniques with switched gradients can be achieved with arbitrary k-space trajectories, including echo-planar (EPSI) [34], spiral [35], radial [36], and concentric ring [37] trajectories, all of which reduce the number of excitations and thus the scan time compared to phase-encoded CSI. However, the resulting speed advantage with fast spectroscopic imaging results in tradeoffs due to gradient limitations between spectral bandwidth, spatial resolution, and SNR efficiency [37].

The fast MRSI approaches form the backbone of several important pilot studies of hyperpolarized ${}^{13}$C in cancer patients. This includes malignancies such as brain cancer [38,39], primary [40,41,42] and metastatic [43] prostate cancer, renal cell carcinoma [44], and pancreatic adenocarcinoma [45]. The ability to cover a continuous chemical-shift spectrum allows resolution of downstream metabolic products without a priori knowledge of their chemical shifts. Such chemical resolution is essential for pilot patient studies investigating new diseases or organs of interest, drug targets and metabolic pathway inhibition, or in the setting of HP probe development.

A potential limitation of fast spectroscopic imaging is that it still requires relatively long scan times owing to the need to explicitly encode the spectral dimension. This limitation can be partially offset by migrating to a multi-echo model-based or imaging-based acquisition strategy (see the following sections) in later phases of a study where the ${}^{13}$C substrate and products are assigned, and main magnetic field variations have been better characterized for the target of interest. Having said this, the fast MRSI approaches still retain important applications in those scenarios where quantitative accuracy and microenvironment characterization have priority over spatial coverage, for probe development when the metabolites are not yet known, or for when spatial localization is limited by the receive profile of the surface coil.

#### 2.2.2. Model-Based Chemical Shift Encoding

Model-based chemical shift encoding techniques, also known as Dixon methods, were originally developed for fat and water MRI and have been successfully adapted to HP ${}^{13}$C MRI to provide faster spectral encoding and shorter scan times when metabolite chemical shifts and the $B_{0}$ field map are known. For the IDEAL CSI technique[46,47], if there are *n* metabolites present, then typical imaging readouts are acquired with at least n+1 different echo times. These can then be reconstructed via matrix inversion into individual metabolite images as well as a field map. An example of this for fast spiral imaging readouts is shown in Figure 3. The efficiency of this approach can also be improved by using over-sampled spiral readouts to reduce the number of echoes required to resolve *n* metabolites [48].

This approach is robust in the sense that it only requires the relative chemical shift of each resonance, and it is fast and readily combined with existing readout techniques to produce a high-resolution final image. Model-based approaches have been used preclinically in the study of kidney [49,50], cardiac [51], and tumor metabolism [47,52] and in healthy human volunteers [53] to characterize brain metabolism. While relatively robust, model-based approaches have their limitations: the multi-echo readout inherently reduces temporal resolution compared to metabolite-selective imaging (see next section), the echo times must be carefully chosen to ensure there is minimal noise amplification from the reconstruction [46], and motion or frequency shifts can make the reconstruction numerically ill-conditioned.

#### 2.2.3. Metabolite-Selective Imaging

Metabolite-selective (also referred to as metabolite-specific) imaging is an extremely fast technique popular for HP ${}^{13}$C MRI that utilizes spectral-spatial (SPSP) RF pulses and a rapid imaging readout to encode the spectral and spatial domains in two distinct steps [54]. Spectral encoding is accomplished by exciting individual resonance frequencies with a spectral-spatial excitation, which is a specialized 2D RF pulse that is both slice- and frequency-selective. Because the spectral encoding is performed with the excitation, conventional MRI imaging readouts including fast single-shot techniques such as spiral or echoplanar trajectories can be employed to rapidly and efficiently encode the magnetization. This approach requires a relatively sparse spectrum with resonances that are known a priori, and is well-suited for studies with [1-${}^{13}$C]pyruvate where all resonances are well-separated (>90 Hz at 3 T).

A typical spectral-spatial RF pulse and schematic of the operation of metabolite-selective imaging can be seen in Figure 4. The singleband spectral-spatial RF pulse performs the spectral encoding, exciting a single metabolite within a slice (or slab) [55]. A rapid imaging readout trajectory, typically single-shot echoplanar [54,56] or spiral [57], is then used to spatially encode the magnetization as a 2D multi-slice or 3D slab encoded dataset. The acquisition then shifts the center frequency and cycles through the resonances of interest over time to acquire a volumetric dynamic dataset for each metabolite. Because each metabolite is excited and encoded separately, multiband flip angle strategies (as described above) that provide metabolite-specific flip angles can be easily integrated, which have been shown to increase SNR over a constant flip angle scheme [56].

It is elucidating to compare the scan time for fast spectroscopic imaging (i.e., EPSI), model-based encoding (i.e., IDEAL), and metabolite-selective imaging (i.e., EPI) sequences. The scan time for a single timeframe is determined by the TR, number of slices (nSlices), and either the number of phase encodes (nPE), number of echoes (nTE), or number of metabolites encoded (nMets):(1)$$\begin{array}{ccc} ScanTime(EPSI) & = & TR\times nPE\times nSlices \end{array}$$(2)$$\begin{array}{ccc} ScanTime(IDEAL) & = & TR\times nTE\times nSlices \end{array}$$(3)$$\begin{array}{ccc} ScanTime(EPI) & = & TR\times nMets\times nSlices \end{array}$$

Given that typically nMets≤nTE<nPE, metabolite-selective imaging provides significant time savings over spectroscopic imaging, typically acquiring data with a temporal resolution of <50 ms/slice/metabolite. This reduction in scan time can be used to increase volumetric coverage or provide data with high temporal resolution, and it has been used extensively in pre-clinical and clinical applications that require high spatiotemporal coverage [58,59]. This sequence is inherently more flexible than spectroscopic imaging, as only the metabolites of interest need to be selectively excited and encoded, reducing the total scan time and eliminating the need to encode the entire spectrum. This approach is therefore well-suited for studies where the hyperpolarized resonances are known a priori and when volumetric coverage and short scan times are required.

However, metabolite-specific imaging is not amenable to all hyperpolarized substrates. Because the SPSP RF pulse performs the spectral encoding, this approach requires a sparse spectrum with well-separated resonances, as the single-shot readouts are unable to resolve multiple chemical species. This in turn is dependent on the chemical shift of metabolites and the operating field strength; as a general rule of thumb, most SPSP RF pulses require at least  90 Hz separation between metabolites to have sufficient stopband suppression and minimize off-resonance excitation. Hyperpolarized substrates such as [1-${}^{13}$C]pyruvate, ${}^{13}$C-bicarbonate, [1,4-${}^{13}$C${}_{2}$]fumarate, and many others meet these criteria.

The SPSP RF pulses and rapid imaging readouts used in metabolite-specific imaging are also more sensitive to center frequency errors and $B_{0}$ field inhomogeneity when compared to fast spectroscopic imaging or model-based imaging approaches. For the imaging gradients, this will manifest as geometric distortion or blurring in metabolite data acquired with echoplanar and spiral readouts, respectively, due to the accumulation of phase during the readout. These artifacts can be partially mitigated by shortening the readout duration, albeit at the cost of reduced SNR efficiency. They can also be reduced through off-resonance and distortion correction strategies developed for 1H MRI that are also compatible with hyperpolarized ${}^{13}$C MRI. For spiral, auto-focus algorithms [57] have been used to correct for $B_{0}$-induced blurring. For EPI, an alternating blip strategy [60] or integrated dual-echo readout [61] can be used to estimate and correct for $B_{0}$ distortion. Symmetric echoplanar readouts also suffer from Nyquist ghost artifacts due to inconsistencies between the phase encodings, which can be corrected for by estimating the phase coefficients from a 1H reference scan using the ${}^{13}$C waveform [56] or via an exhaustive search [62].

Finally, because the SPSP RF pulse performs the spectral encoding, proper frequency calibration is crucial. Miscalibration of the center frequency and $B_{0}$ field inhomogeneity will reduce the applied flip angle due to the narrow passband of the SPSP RF pulse (typically ± 2.5 ppm full-width half-maximum bandwidth for [1-${}^{13}$C]pyruvate applications) and can introduce off-resonance artifacts. With large shifts, this can lead to a failure to excite the metabolites of interest. With smaller shifts, the lower flip angle will lead to overall reduced SNR and can potentially bias quantification if left unaccounted.

#### 2.2.4. Refocused Imaging Methods

The imaging methods discussed above are typically acquired as gradient echo sequences, where transverse magnetization is spoiled at the end of each readout. A more efficient way to use the nonrenewable hyperpolarized magnetization is to refocus transverse spins, which is especially valuable for imaging metabolites with long $T_{2}$ such as [1-${}^{13}$C]pyruvate, [1-${}^{13}$C]lactate, and [${}^{13}$C, ${}^{15}$N${}_{2}$]urea [24,63,64].

In particular, balanced steady-state free precession (bSSFP) can provide high SNR efficiency by taking advantage of the long $T_{2}$ times through repeated pulse trains. In the bSSFP sequence, a train of refocusing pulses is applied with alternating polarity and net gradient areas that are always zero between two neighboring refocusing pulses [65]. This sequence is characterized by an intrinsic periodic frequency response commonly referred to as banding artifacts that appear at repetitions of ±1/2TR.

For angiography and perfusion studies using a metabolically inactive substrate (such as [${}^{13}$C, ${}^{15}$N${}_{2}$]urea or HP001), spectral encoding can be neglected [24,66]. For metabolizable substrates, a multi-echo readout can be used in conjunction with a model-based reconstruction to encode spectral information. Leupold et al. [67] used a symmetric multi-echo readout to quantify renal [1-${}^{13}$C]pyyruvate metabolism in healthy pigs and showed significant SNR improvement over 2D CSI.

Alternatively, a metabolite-selective approach can be performed using spectrally selective RF pulses in a manner analogous to Figure 4 [23,68]. Using a metabolite-selective 3D bSSFP sequence, Tang et al. showed in animal models and healthy human volunteers that bSSFP provides more than a 2.5-fold increase in lactate SNR when compared to a gradient spoiled acquisition [25].

With bSSFP, the choice of flip angle is a tradeoff between alleviating banding artifacts and preserving magnetization for dynamic imaging. A large flip angle (>100${}^{\circ }$) is favorable to reduce banding artifacts but limits the total available scan time because the majority of the magnetization is tipped into the transverse plane, leading to signal decay primarily via $T_{2}$ rather than $T_{1}$ relaxation. However, to perform dynamic imaging in hyperpolarized studies, a small flip angle is necessary to preserve sufficient magnetization for later timepoints. An intermediate flip angle (e.g., 60${}^{\circ }$) can be used to achieve a compromise between the two factors. In addition to TR and flip angle, the spectral response of the RF excitation profile can also influence the bSSFP signal [23] and must also be taken into consideration to avoid off-resonance excitation.

#### 2.2.5. Acceleration Methods

Because of the limited lifetime of hyperpolarized substrates, reducing the total scan time and increasing temporal resolution can improve SNR and image quality. For pre-clinical studies, acceleration can be achieved through undersampling and compressed sensing reconstruction [69] and is particularly useful when multichannel arrays are not available. HP ${}^{13}$C MRI is particularly amenable for compressed sensing because it is typically not limited by SNR but rather by encoding time, and the resulting spectra are typically sparse.

Compressed sensing has been demonstrated in 2D metabolic imaging of the rat heart using a single-shot echo-planar readout at a spatial resolution of 1 × 1 × 3.5 mm${}^{3}$ with up to 5-fold acceleration [70] and in rat kidneys with a 3D EPI sequence using a psueudorandom blip scheme in the z-dimension, achieving up to 3-fold acceleration [71]. To overcome the bandwidth limitations of EPSI, blip gradients can be applied during the readout in phase-encoded 2D MRSI for both acceleration and high-bandwidth applications [72]. Compressed sensing can also be used in conjunction with model-based approaches to further accelerate the acquisition [49,70] and can be extended to dynamic imaging using incoherent sampling and low-rank reconstruction to improve the spatial and/or temporal resolution [73]. However, choosing the proper value for the regularization parameter requires numerical simulations or empirical studies to determine the tradeoff between data consistency and excessive denoising, which can impact quantification if not properly tuned.

#### 2.2.6. Calibration Methods

Accurate RF calibration and variations in bolus timing between studies present significant challenges to acquiring robust and reproducible hyperpolarized datasets. Center frequency and RF power calibration is crucial to the success of all imaging approaches but is difficult to perform due to the lack of endogenous ${}^{13}$C signal. One common strategy is to perform this calibration on an external ${}^{13}$C-enriched phantom [42,53,74], but this approach does not fully account for $B_{0}$ or $B_{1}^{+}$ inhomogeneity throughout the field of view. As an alternative, center frequency calibration can be performed based on the measured water 1H frequency and then converted to the ${}^{13}$C operation frequency after accounting for differences in chemical shift [75]. For improved RF power calibration, Grist et al. recently showed that the more abundant ${}^{23}$Na resonance can be used to calibrate the ${}^{13}$C $B_{1}^{+}$ field with high reproduciblity [75].

Appropriate acquisition timing for hyperpolarized ${}^{13}$C imaging is also important to avoid saturation of the non-recoverable hyperpolarized spins during bolus arrival, particularly near the transmitter conductive elements where the $B_{1}^{+}$ is high. Many variable flip angle schemes [15,17,18,19,76] also assume knowledge of the bolus shape and timing, but variability in bolus kinetics leads to quantification errors in the context of ratiometric measures [77,78,79]. A fixed delay between injection and acquisition has been employed in many hyperpolarized ${}^{13}$C imaging studies [32,40,53,80,81], but the empirically determined delay time can be unreliable due to the inherent physiological variability between subjects, and in human cancers where the vascularization and perfusion are highly variable over subjects [82]. This is particularly problematic in human subjects where timing differences of up to 12 seconds have been observed in hyperpolarized ${}^{13}$C studies of prostate cancer patients [40,41].

Variable perfusion characteristics can be resolved using a bolus tracking technique to trigger the acquisition upon pyruvate arrival, as demonstrated in work by Durst et al. [77] and Tang et al. [83] on clinical scanners and by Blazey et al. on a pre-clinical system [84]. Center frequency and RF power calibration using a Bloch–Siegert approach [85] can also be incorporated into bolus tracking [83], eliminating the challenges associated with RF calibration and variations in bolus timing. The primary disadvantage of using a real-time bolus tracking framework is the engineering efforts to address system compatibility and the potentially complicated analysis of bolus kinetics.

## 3. Data Analysis Methods

Quantification and analysis of HP ${}^{13}$C MRI data requires specialized methods that account for the metabolic conversion, real-time kinetics, and variations in signal amplitude (e.g., due to polarization, RF coils, perfusion, and substrate delivery). When using dynamic acquisitions, pharmacokinetic models have proven to be a powerful method because they can reflect the dynamics of an HP ${}^{13}$C MRI experiment. Model-free metrics are also often used, and when used properly can also provide accurate measures of metabolic conversion. Both kinetic modeling and model-free metrics are discussed in this review.

### 3.1. Kinetic Modeling

Pharmacokinetic (PK) models have been widely applied to HP ${}^{13}$C MRI because they provide a clear framework for capturing the metabolite dynamics. At a minimum, these models should account for metabolic conversion/flux. They typically also include relaxation and RF pulse effects, and can also be extended to included perfusion and cellular transport. However, it should also be recognized that even the best and most complete PK model is an approximation, and that it is possible to generate models that are excessively complex and that may contain more unknowable parameters than our observed data can support.

Most commonly, a precursor-product model is used to describe the relationship between the longitudinal magnetization of the injected substrate, S(t), and metabolic products, M(t) (e.g., most commonly ${}^{13}$C-pyruvate as the substrate and ${}^{13}$C-lactate as a product):(4)$$\frac{d}{dt}\left[\begin{array}{c} S(t)\\ P(t) \end{array}\right]=\left[\begin{array}{cc} -R_{S}-k_{S,P} & +k_{P,S}\\ +k_{S,P} & -R_{P}-k_{P,S} \end{array}\right]\left[\begin{array}{c} S(t)\\ P(t) \end{array}\right]+\left[\begin{array}{c} u(t)\\ 0 \end{array}\right]$$
where $R_{S},R_{P}$ are the longitudinal relaxation rate ($1/T_{1}$), $k_{S,P}$ and $k_{P,S}$ are the conversion rates from substrate-to-product and product-to-substrate, respectively, and u(t) is an optional input function to model new substrate magnetization. This input function can be fit to a boxcar [86], gamma-variate [17], or other shape, or can be estimated based on the data [21,87]. In many cases, including for HP ${}^{13}$C-pyruvate, the product-to-substrate conversion rate is assumed to be negligible, i.e., $k_{P,S}=0$ during the relatively short HP experiment.

This model, however, typically will underestimate the metabolic conversion rate since there is usually a large amount of substrate magnetization in the vasculature during the short imaging window after bolus injection. This can be addressed by separating the metabolite signals into two physical compartments: intravascular ${}_{iv}$ and extravascular ${}_{ev}$ spaces (middle in Figure 5) [88]. The kinetic model then becomes:(5)$$\frac{d}{dt}\left[\begin{array}{c} S_{ev}\left(t\right)\\ P_{ev}\left(t\right) \end{array}\right]=\left[\begin{array}{cc} -R_{S}-k_{S,P}-\frac{k_{ve}}{v_{e}} & +k_{P,S}\\ +k_{S,P} & -R_{P}-k_{P,S}-\frac{k_{ve}}{v_{e}} \end{array}\right]\left[\begin{array}{c} S_{ev}\left(t\right)\\ P_{ev}\left(t\right) \end{array}\right]+\frac{k_{ve}}{v_{e}}\left[\begin{array}{c} S_{iv}\left(t\right)\\ P_{iv}\left(t\right) \end{array}\right]$$
which now includes the extravasation rate, $k_{ve}$, and extracellular volume fraction, $v_{e}$. The substrate intravascular magnetization, $S_{iv}\left(t\right)$, can be estimated from a vascular input function measurement, usually from a separately identified vascular voxel, and the production intravascular magnetization is typically assumed to be zero, $P_{iv}\left(t\right)=0$. There is flexibility in the accuracy of estimates of $k_{ve}$, $v_{e}$, and $S_{iv}\left(t\right)$ [78].

This model can be further adapted to also include separate extravascular-extracellular and intracellular physical compartments, although analyses show that HP in vivo data typically do not support this level of model complexity [88].

RF pulse effects must also be accounted for in kinetic modeling, since the above differential equations only describe the behavior of the magnetization in the absence of RF pulses. This can be accomplished by using a hybrid discrete-continuous model in which the RF pulses are a discrete event (rotation) along with the continuous signal evolution described above [17]. The RF pulses are modeled as
(6)$$\begin{array}{ccc} X_{Z}^{+}\left[n\right] & = & X_{Z}^{-}\left[n\right]\cos\theta_{X}\left[n\right] \end{array}$$
(7)$$\begin{array}{ccc} X_{S}\left[n\right] & = & X_{Z}^{-}\left[n\right]\sin\theta_{X}\left[n\right] \end{array}$$
where $X_{Z}^{-}\left[n\right]$ and $X_{Z}^{+}\left[n\right]$ are the longitudinal magnetization for metabolite *X* before and after the $n^{th}$ RF pulse, respectively, $X_{S}\left[n\right]$ is the signal (e.g., transverse magnetization) for metabolite *X* after the $n^{th}$ RF pulse, and $\theta_{X}\left[n\right]$ is flip angle for metabolite *X* for the $n^{th}$ RF pulse.

These models can be solved through non-linear least-squares curve-fitting methods. The hyperpolarized-mri-toolbox (https://github.com/LarsonLab/hyperpolarized-mri-toolbox, accessed on 11 June 2021) [89] provides implementations of several of these kinetic models and examples including application to in vivo data. These models can be extended to include additional metabolic products (e.g., [1-${}^{13}$C]-lactate, [1-${}^{13}$C]-alanine, and ${}^{13}$C-bicarbonate as products following [1-${}^{13}$C]-pyruvate substrate injection), and this capability is built into this toolbox.

### 3.2. Model-Free Metrics

Kinetic models are conceptually appealing in that they attempt to quantify relationships between biological compartments and biochemical processes, with capability to improve the specificity to which potential changes could be attributed with increasing model complexity. Furthermore, the kinetic rates should be invariant to different experimental conditions (e.g., flip angles, temporal resolution) for comparisons across experiments. However, they have several disadvantages: simpler kinetic models are often favored due to practical limitations of the HP experiment; other information that may be difficult to obtain (e.g., compartment volumes, vascular input functions) but are needed for more complex models; the results are susceptible to errors when there is mismatch between the model and the data; instabilities in model fitting, particularly with more complex models; and they require more sophisticated implementations than many of the model-free alternatives described next. For these reasons, model-free metrics of metabolic conversion, several of which are illustrated in Figure 6, have been developed. They are a popular alternative to kinetic models that can be applied effectively based on limitations described below.

The most common model-free metrics used in quantification of HP data are metabolite ratios in which the denominator is the substrate or total ${}^{13}$C signal. These “Ratiometric” approaches, three of which are described below, have several advantages. First and foremost, they provide a metric reflecting metabolic conversion that is straightforward to compute; they also include inherent normalization such that they are largely invariant to the amount of polarization, RF coil sensitivities, and amount of agent delivered.

#### 3.2.1. Single time-point Metabolite Ratio

**Single time-point**: In the simplest form, metabolite ratios can be computed from data at a single time-point, as illustrated in Figure 6 by $Lac\left(t_{meas}\right)/Pyr\left(t_{meas}\right)$. However, this approach is very sensitive to the timing of this single time-point because of the typically rapid rates of bolus delivery, metabolic conversion, and signal decay, which results in signals and the ratio between them varying continuously throughout their observable lifetime. This can be partially addressed by selecting a stable and reproducible measurement time relative to the bolus in a dynamic acquisition [41], if one can be identified.

#### 3.2.2. Area-under-curve (AUC) Metabolite Ratio

The more robust approach is to perform a dynamic acquisition and compute the area-under-curve (AUC) ratio from the metabolite time courses, illustrated in Figure 6 by $AUC_{ratio}$. This approach has been shown to be proportional to the metabolic conversion rate under certain conditions [91]. With assumptions that the acquisition covers the entire dynamic acquisition, including the bolus delivery, and an underlying precursor-product kinetic model (Equation (Equation 4), left on Figure 5), the AUC ratio between lactate and pyruvate is the ratio of the AUC for each metabolite:(8)$$AUC_{ratio}=\frac{AUC_{Lac}}{AUC_{Pyr}}=\frac{k_{PL}}{R_{Lac}^{\prime }+k_{LP}}$$
where $k_{PL},k_{LP}$ are the conversion rate from pyruvate-to-lactate and lactate-to-pyruvate, respectively, and $R_{Lac}^{\prime }$ is the effective decay rate of lactate that includes $T_{1}$ and RF pulse losses.

This relationship trivially extends to other HP agents with precursor-product conversion processes, and also it is valid when the substrate is converted into multiple products (e.g., [1-${}^{13}$C]pyruvate to [1-${}^{13}$C]alanine and ${}^{13}$C-bicarbonate).

This proportionality holds for all flip angle schemes that do not change over time. It does, however, depend on the flip angles used ($R_{Lac}^{\prime }$ includes excitation losses), meaning the $AUC_{ratio}$ cannot be compared across studies that use different flip angles without additional corrections. The proportionality between the $AUC_{ratio}$ and $k_{PL}$ in Equation (Equation 8) breaks down when flip angles vary over time and if there are any variations in substrate delivery timing.

In some cases, the differential metabolic conversion is informative. For example, when studying glucose metabolism versus oxidative metabolism, the signal amplitude or $AUC_{ratio}=AUC_{bicarbonate}/AUC_{lactate}$ between metabolic products following [1-${}^{13}$C]pyruvate injection will reflect differences in metabolic preference. Because both products depend on substrate delivery in the same way, differential conversion may be relatively insensitive to vascular or bolus characteristics.

## 4. Final Remarks

This review described state-of-the-art methods for acquisition, reconstruction, and analysis of data for HP ${}^{13}$C MR experiments. For RF flip angle strategies, they can be designed to enable dynamic imaging by preserving magnetization with smaller flip angles, and in particular using small flip angles for the HP substrate preserves magnetization for metabolic conversion. For data acquisition, slice-localized MRS provides high SNR and robust performance, MRSI adds on additional localization while preserving full spectral resolution, and fast imaging techniques such as IDEAL and metabolite-specific imaging use prior knowledge of the HP spectrum to provide rapid, volumetric coverage in vivo. For data analysis, kinetic models allow for quantification of metabolic conversion independent of the acquisition strategies, while model-free metrics such as area-under-curve ratios are robust to calculate but will vary based on pulse sequence parameters. Taken together, we hope this review provides helpful guidance for designing and analyzing HP ${}^{13}$C MRI experiments.

## Figures and Tables

**Figure 1 metabolites-11-00386-f001:**
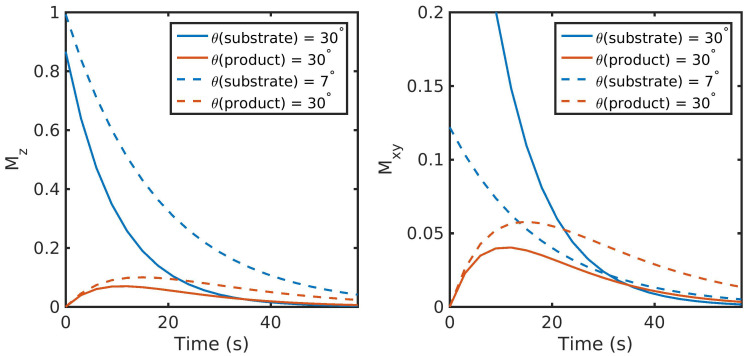
Illustration of two flip angle schemes for HP MRI and their resulting impact on the longitudinal (Mz) and transverse (Mxy) magnetization. Utilizing the same flip angle for both substrate and product results in excess substrate SNR and an overall reduction in product SNR because of the rapid drop in substrate Mz. Using a metabolite-specific flip angle scheme—lower flip angle for substrate and higher for the metabolic product—results in reduced usage of the substrate hyperpolarization, providing 1.8-fold more magnetization for the product in this example while still providing sufficient substrate SNR throughout the acquisition. Data were simulated with a precursor-product kinetic model shown in Figure 5 and described in Equation (Equation 4), initialized with all magnetization in the substrate and no additional input function. Other parameters were: 3 s TR, a $T_{1}$ of 30 s for both substrate and product, and metabolic conversion rate constants of $k_{S,P}=0.02$ s${}^{-1},k_{P,S}=0$.

**Figure 2 metabolites-11-00386-f002:**
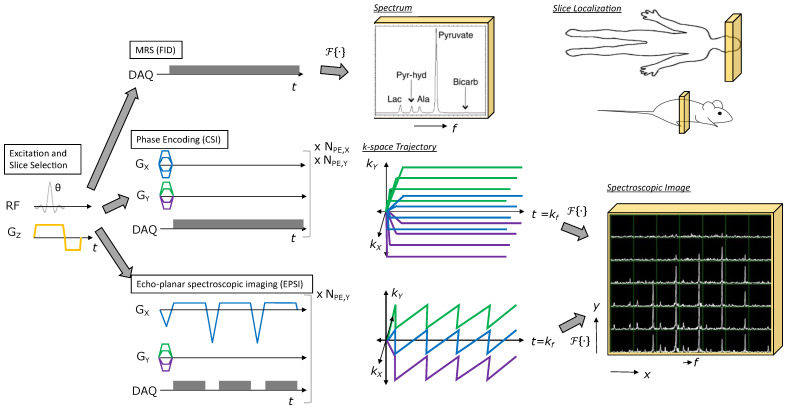
Illustration of MRSI methods for HP agents. All methods start with RF excitation and slice selection, appended with a spectroscopic or spectroscopic imaging readout. Free induction decay (FID) MRS provides a spectrum from the excited slice and just requires one TR. Phase encoding (CSI) provides a spectroscopic image, but requires multiple TRs to perform all phase encodings necessary to sample k-space (e.g., Scan time = $TR\times N_{PE,X}\times N_{PE,Y}$ for the 2D MRSI example shown). Echo-planar spectroscopic imaging (EPSI) also provides a spectroscopic image, but requires relatively fewer TRs to cover k-space (e.g., Scan time = $TR\times N_{PE,Y}$ for the 2D MRSI example shown), allowing for rapid imaging of HP agent kinetics.

**Figure 3 metabolites-11-00386-f003:**
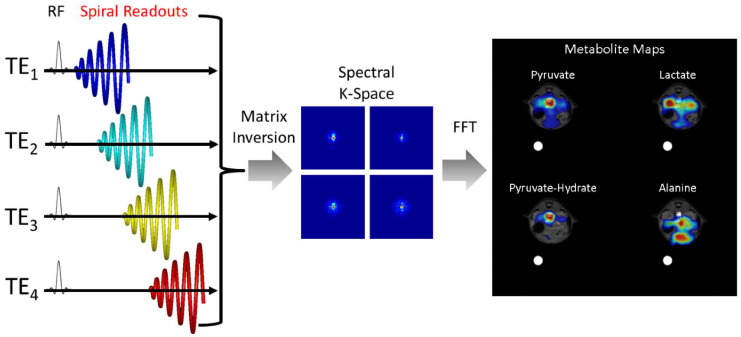
Schematic illustrating the acquisition and reconstruction of a model-based spiral pulse sequence. In this illustration, data are acquired with a long duration spiral readout that is spoiled at each TR. Each subsequent excitation is shifted in time by ΔTE. For non-Cartesian approaches, a matrix decomposition occurs in k-space, yielding k-space data for each metabolite. A subsequent gridding or nonuniform fast Fourier Transform (nuFFT) step yields spectral images for each hyperpolarized metabolite.

**Figure 4 metabolites-11-00386-f004:**
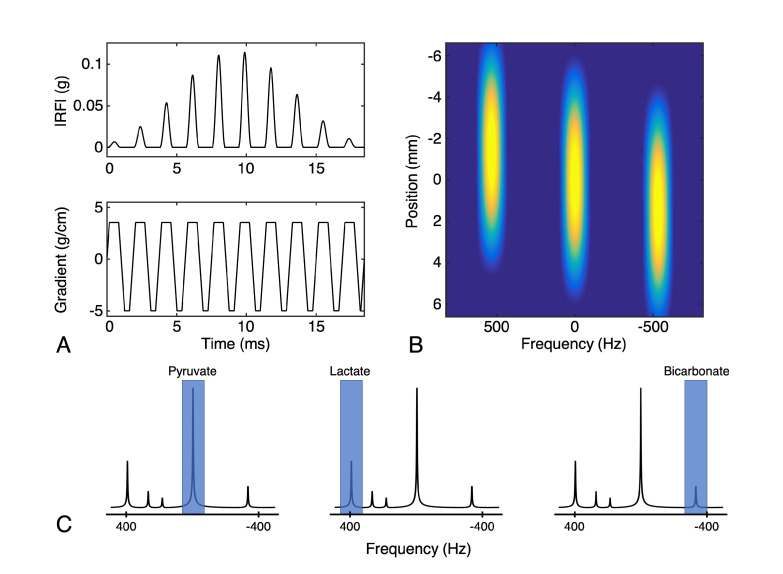
Illustration of the metabolite-selective imaging technique. (**A**) Example of a spectral-spatial RF pulse designed for studies of HP [1-${}^{13}$C]pyruvate and [1-${}^{13}$C]lactate. The RF envelope and oscillating gradient provide the spectral and spatial selectivity of the 2D RF pulse. (**B**) The transverse magnetization (Mxy) excited by this pulse shows the narrow passband frequency response needed for metabolite-selective imaging as well as slice-selectivity for spatial localization. (**C**) The singleband spectral-spatial RF pulse performs the spectral encoding by exciting a single metabolite within a narrow passband (highlighted in blue) over a single slice (or slab). A rapid imaging readout trajectory is then used to spatially encode the magnetization as a 2D multi-slice or 3D phase-encoded dataset. The acquisition then shifts the center frequency and cycles through the resonances of interest (pyruvate, lactate, and bicarbonate at 3T in this example) over time to acquire dynamic data for each metabolite.

**Figure 5 metabolites-11-00386-f005:**
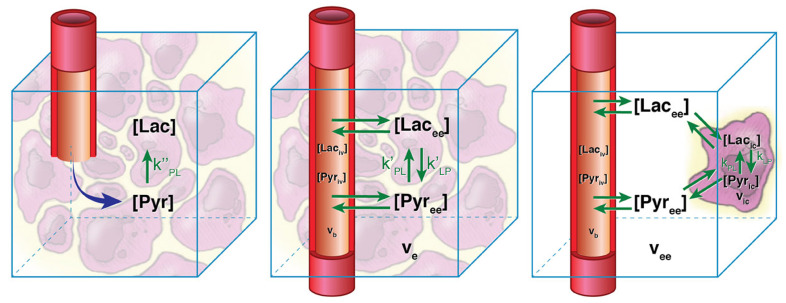
Illustration of kinetic models with increasing complexity, starting with the precursor-product model (**left**), separation of intravascular and extravascular physical compartments (**middle**), and further separation of an intracellular compartment (**right**). Precursor-product models (**left**) are the easiest to apply, but two physical compartment models (**middle**) have been shown to be more appropriate for in vivo HP data and will more accurately estimate metabolic conversion rates without vascular contamination. Figure adapted from Ref. [88].

**Figure 6 metabolites-11-00386-f006:**
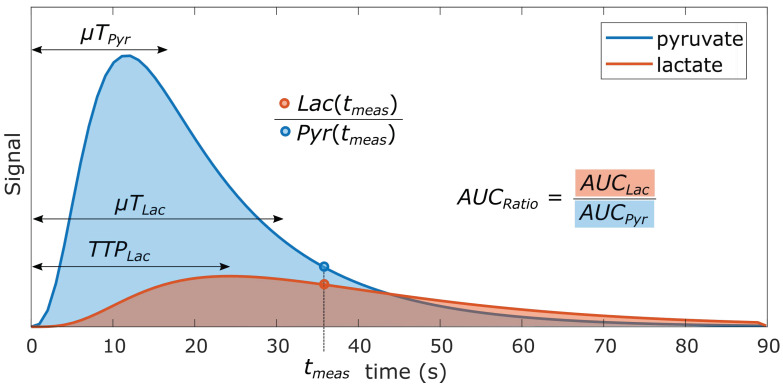
Illustration of several model-free metrics that can be used for quantifying HP data. These include the lactate-to-pyruvate ratio at a single time-point (Lac(tmeas)/Pyr(tmeas)), the area under the curve ratio (AUCratio), lactate time-to-peak (TTPLac), and the mean time (μT) for a metabolite. Inspired by [90].

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
