# Peer review of "Hyperpolarized Metabolic MRI—Acquisition, Reconstruction, and Analysis Methods"

_metabolites, 2021, doi:10.3390/metabo11060386_

Round 1
Reviewer 1 Report
Authors of this review article have discussed details about the state of the art of hyperpolarized MRI in terms of data acquisition, reconstruction and analysis methods. The article is well written will required technical information.
Some points to consider :
The authors should describe some limitations and challenges regarding the lifetime of molecules that are available for pre-clinical and clinical applications. Discuss about the polarization efficiency of molecules that will be suitable for clinical translation. Introduce a table highlighting various molecules that are currently available. Include the relaxation times at clinical / pre-clinical field strengths.
Page 2 Line 48: This is a known fact about the increase in the SNR but please include few references .
Elaborate more more details about the Fig 1 and their simulation methodology.
Page 4: Lines 135-137 authors should expand the legend of the Fig 2 to explain more details
The authors should also include a section on emerging techniques with Long Lived Spin States (Work of Malcom Levitt, Bodenhausen and others)
Author Response
R1.1 The authors should describe some limitations and challenges regarding the lifetime of molecules that are available for pre-clinical and clinical applications. Discuss about the polarization efficiency of molecules that will be suitable for clinical translation. Introduce a table highlighting various molecules that are currently available. Include the relaxation times at clinical / pre-clinical field strengths.
We have expanded the introduction to comment on the range of T1 values, including changes with magnetic field and also relaxation mechanisms. We have also added references that provide greater details on agents for HP studies.
R1.2 Page 2 Line 48: This is a known fact about the increase in the SNR but please include few references .
We have included a reference to the seminal paper by Ardenkjaer-Larsen et al about this polarization increase.
R1.3 Elaborate more more details about the Fig 1 and their simulation methodology.
We have added additional details about the signal model used to the caption.
R1.4 Page 4: Lines 135-137 authors should expand the legend of the Fig 2 to explain more details
In response to this comment and request from reviewer 2, we have revised Fig 2 and updated the caption to provide additional detail.
R1.5 The authors should also include a section on emerging techniques with Long Lived Spin States (Work of Malcom Levitt, Bodenhausen and others)
We have included a reference to a recent review on long lived spin states to highlight this potentially transformative improvement.
Reviewer 2 Report
This is a very detailed and carefully prepared manuscript, which provides an in-depth review of acquisition, reconstruction and analysis methods in the important field of hyperpolarized MRI with 13C-labeled agents. 13C-imaging is one of the few and rapidly developing MRI-based approaches to probe important metabolic processes, and this review is a timely effort to systematize the classic and most recent technological developments in the area. I have just few minor comments:
- Though the manuscript is generally well written, it would benefit from more polishing. For example, Section 2.2.1 (Magnetic resonance spectroscopic imaging) needs substantial style and grammar work, especially its first paragraph. For example, the first sentence is constructed incorrectly, while the second one contains too many clauses making it hard to comprehend.
- Figure 2 may be confusing to less experienced readers. Please consider redesigning it to make the content in general and concept of kf encoding in particular easier to understand.
- While the scope of reference is almost complete, it is not clear why the authors did not cite advanced spiral acquisition/model-based reconstruction technique of Magn Reson Med. 2014 Apr;71(4):1435.
Author Response
R2.1 Though the manuscript is generally well written, it would benefit from more polishing. For example, Section 2.2.1 (Magnetic resonance spectroscopic imaging) needs substantial style and grammar work, especially its first paragraph. For example, the first sentence is constructed incorrectly, while the second one contains too many clauses making it hard to comprehend.
We apologize for this lack of polishing. We have proofed the manuscript and made style updates throughout.
R2.2 Figure 2 may be confusing to less experienced readers. Please consider redesigning it to make the content in general and concept of kf encoding in particular easier to understand.
We have redesigned Fig.2 and the associated caption with the goal of clarity and simplification.
R2.3 While the scope of reference is almost complete, it is not clear why the authors did not cite advanced spiral acquisition/model-based reconstruction technique of Magn Reson Med. 2014 Apr;71(4):1435.
This reference has been added.